# Carbon Footprint Management by Agricultural Practices

**DOI:** 10.3390/biology11101453

**Published:** 2022-10-02

**Authors:** Ekrem Ozlu, Francisco Javier Arriaga, Serdar Bilen, Gafur Gozukara, Emre Babur

**Affiliations:** 1Vernon G. James Research Center-Tidewater Research Station, Department of Crop and Soil Sciences, North Carolina State University, 207 Research Station, Plymouth, NC 27962, USA; 2Department of Soil Science, University of Wisconsin-Madison, Madison, WI 53705, USA; 3Department of Soil Science and Plant Nutrition, Faculty of Agriculture, Ataturk University, Erzurum 25100, Turkey; 4Department of Soil Science and Plant Nutrition, Faculty of Agriculture, Eskisehir Osmangazi University, Eskisehir 26250, Turkey; 5Department of Soil Science and Ecology, Faculty of Forestry, Kahramanmaraş Sütçü İmam University, Kahramanmaraş 46050, Turkey; 6Department of Biochemistry and Microbiology, School of Environmental and Biological Sciences, Rutgers, The State University of New Jersey, New Brunswick, NJ 08901, USA

**Keywords:** carbon footprint, agriculture, soil management, carbon sequestration, greenhouse gas emissions

## Abstract

**Simple Summary:**

Minimizing the effects of climate change by reducing GHG emissions is crucial and can be accomplished by truly understanding the carbon footprint phenomenon. This study aims to improve the understanding of carbon footprint alteration due to agricultural management and fertility practices. It provides a detailed review of carbon footprint management under the impacts of environmental factors, land use, and agricultural practices. The results show that healthy soils have numerous benefits for the general public and especially farmers. These benefits include being stable and resilient, resistant to erosion, easily workable in cultivated systems, good habitat for soil micro-organisms, fertile and good structure, large carbon sinks, and hence lower carbon footprint. Intensive tillage is harmful to soil structure by oxidizing carbon and causing GHG emissions. If possible, no-till; if not, minimum tillage frequency and depth of tillage, and optimum moisture are recommended. The soil should be at an appropriate level of moisture when tillage takes place. Diverse cropping systems are better for the soil than monocultures. Minimizing machinery operations can help to avoid soil compaction. Building soil organic carbon in the most stable form is the most efficient practice of sustainable crop production.

**Abstract:**

Global attention to climate change issues, especially air temperature changes, has drastically increased over the last half-century. Along with population growth, greater surface temperature, and higher greenhouse gas (GHG) emissions, there are growing concerns for ecosystem sustainability and other human existence on earth. The contribution of agriculture to GHG emissions indicates a level of 18% of total GHGs, mainly from carbon dioxide (CO_2_), methane (CH_4_), and nitrous oxide (N_2_O). Thus, minimizing the effects of climate change by reducing GHG emissions is crucial and can be accomplished by truly understanding the carbon footprint (CF) phenomenon. Therefore, the purposes of this study were to improve understanding of CF alteration due to agricultural management and fertility practices. CF is a popular concept in agro-environmental sciences due to its role in the environmental impact assessments related to alternative solutions and global climate change. Soil moisture content, soil temperature, porosity, and water-filled pore space are some of the soil properties directly related to GHG emissions. These properties raise the role of soil structure and soil health in the CF approach. These properties and GHG emissions are also affected by different land-use changes, soil types, and agricultural management practices. Soil management practices globally have the potential to alter atmospheric GHG emissions. Therefore, the relations between photosynthesis and GHG emissions as impacted by agricultural management practices, especially focusing on soil and related systems, must be considered. We conclude that environmental factors, land use, and agricultural practices should be considered in the management of CF when maximizing crop productivity.

## 1. Introduction

The debate about the anthropogenic impact of climate change on a global scale has been increasing over the last 50 years as the detrimental impact of increased temperatures is now widely recognized by the scientific and the non-scientific communities at the same time. Earth surface temperatures are expected to rise between 1.6 to 5.8 °C by end of the century, parallel to the population growth rate and greenhouse gas (GHG) emissions [1]. With 95% confidence, the anthropogenic impact in natural agroecosystems has been found responsible for the mainstream climate change [2] observed since the late 1800s. Therefore, decreasing these GHG emissions to the atmosphere is an important task that might be achieved through a keen understanding of the carbon footprint (CF) of human activities. Besides quantifying total GHG impacts, knowledge of the CF can provide a strategy with an inclusive GHG record to distinguish expected reductions from its major sources. Thus, CF calculations might be a successful tool to guide the reduction in emissions and understand the risk of global warming. The term ‘carbon footprint’, which has become extremely common and is now a worldwide concept [3] was defined as “a measurement of the total GHG emissions caused directly or indirectly by an individual, an organization, even a product, and is expressed as a carbon dioxide equivalent (CO_2_e)” by Gao, et al. [4]. As such, CF is a weight in units of kg or Mg of carbon per person or activity. Several methods have been allocated to determine estimates of CF, ranging from basic online tools to complex models, life-cycle analysis, or input-output-based methods and tools [3].

The global food and agricultural industry is challenged by various issues, among the most important of which is the necessity for the industry to produce a higher quality and a sufficient food supply that cater for the increasing global population. At the same time, the main environmental conditions within agricultural production and animal husbandry are heavily scrutinized by the public and governmental and non-governmental organizations. Previous attempts have indicated that 70% to 90% of the environmental impacts of agricultural production occur during the primary production process [5], which further indicates the importance of CF as part of the ecological footprint.

Supply and energy-intensive agricultural practices were reported to have high CF which has increased the global energy budget by about 10 times since the beginning of the 20th century [6]. Atmospheric CO_2_ concentration has been increasing dramatically since the beginning of the industrial revolution, when the estimated CO_2_ concentration was about 310 ppm, compared to the current atmospheric CO_2_ concentration of 418 ppm measured on 1 July 2021, according to the Keeling Curve (https://keelingcurve.ucsd.edu/ (accessed on 1 July 2021)). The Kyoto Protocol established an obligation to mitigate the increases in GHG emissions to 108% of its 1990 records by 2012 [7]. Besides the stationary energy, industrial, and transport sectors, the agricultural sector must be strongly committed to applying its tools to diminish GHGs and climate change [7]. China [10151 million metric tons (MMT) CO_2_] and the USA (5312 MMT CO_2_) have been reported as the leading countries in terms of atmospheric CO_2_ emissions, followed by India (2431 MMT CO_2_), Russia (1635 MMT CO_2_), and Japan (1209 MMT CO_2_). However, in terms of methane (CH_4_) emissions, tropical South America and Southeast Asia are the global leading regions, followed by China, Central Eurasia, Japan, and Southern Africa. Most of the CH_4_ emissions on a global scale are related to agricultural and waste activities (191 MMT CH_4_ yr^−1^) or wetlands (167 MMT CH_4_ yr^−1^). In particular, India and China are the countries to have the most CH_4_ emissions because of agriculture and waste. However, most CH_4_ emissions come from wetlands in Tropical South America. The Intergovernmental Panel on Climate Change (IPCC) reported that GHG emissions as a result of waste management were around 3% of the total emissions in 2010 [8].

As part of agriculture and waste activities, rice is responsible for a significant amount of CH_4_ emissions, especially in India and China. Similarly, CH_4_ and nitrous oxide (N_2_O) emissions from agricultural lands represent 59% and 84% of the total CH_4_ and N_2_O gas emissions in Australia, respectively [9]. In Australia, livestock is the largest source of emissions as a part of the agricultural sector, producing about 62.8 MMT CO_2_, which represents almost 70% of the total agricultural GHG emissions [7]. Measuring the impact of the dairy industry is complex and varies with climate and management practices which influences not only the GHG emission per kg of energy-corrected milk or meat invention but also the runoff of phosphorus (P) from agricultural fields.

The agricultural sector has recently advocated switching chemical sources with organic active elements to biological or biodynamic growing methods. Using chemical fertilizers in agriculture has been shown to increase GHG emissions, especially N_2_O, thus contributing to enhancing climate change issues [10]. Therefore, agricultural practices need to be reviewed and readapted to be environmentally friendly. Following, the CF concept is discussed under both mechanistic and practical approaches to assessing the contribution of agricultural activities to global sustainability. Therefore, the purposes of this study were to improve understanding of CF alteration due to agricultural management and fertility practices.

## 2. Carbon Footprint Due to Environmental Factors

Carbon footprint (CF) is a popular element in the agro-environmental sciences due to its role in the environmental impact assessments related to alternative solutions and global climate change. The International Organization for Standardization (ISO) 14001 certification is a starting point of prospectus obligations for concerns in environmental health. Tools such as CF have achieved increasing interest for the recognition of international standards. For this, using available tools and matching international standards need a balance between economic and environmental elements. Therefore, interest in monitoring carbon (C) loss through GHG emissions and C sequestration from/to agricultural and non-agricultural fields has been on the rise.

Supra-optimum temperatures negatively affect crop growth, cause deforestation, and change vegetation patterns that can cause lower photosynthesis and increase CO_2_ concentrations in the atmosphere. Soil properties such as soil temperature, soil moisture, water-filled pore space (WFPS), and soil micro-ecology can also influence CF [11]. The temperature was documented as a major driver of CO_2_ fluxes and microbial activity [10].

Soil moisture content is an essential soil property due to its impact on soil gas emissions and regulation of microbial activities and related processes [12]. For instance, nitrifying bacteria need oxygen in soil pores [13]. Therefore, soils with lower moisture content provide more GHG emissions associated with nitrification which causes a higher potential for nitric oxide (NO) emissions than N_2_O [14]. However, lower than 10% WFPS can result in lower NO emissions due to reserved nutrient sources [15]. In addition, an anaerobic environment is needed for CH_4_ and N_2_O-producing bacteria [13], which is positively correlated with soil moisture [16].

Soil temperature is also an important component due to its influence on the variations of soil GHG emissions. Alongside the soil moisture, temperature changes can explain a significant portion of the variability in both NO (74%) and N_2_O (86%) emissions [17]. If other conditions remain similar, increases in soil temperature result in higher emissions. Higher soil respiration rates are positively associated with microbial activities but negatively associated with soil O_2_ content [13]. Even though interactions between moisture and temperature effects occur at the same time under field conditions [18], GHG emissions generally increase from winter to summer and decrease from summer to winter due to changes in both temperature and soil moisture contents (Ozlu and Kumar, 2018b). Other factors are somehow related to temperature and moisture content in soils. For example, the exposure impacts soil temperature and moisture [13]. Lower air pressure causes higher GHG emissions owing to lower counter pressure on the soil. For instance, N_2_O is higher in depressions than on slopes and ridges because of increases in soil moisture [19]. Similarly, burning vegetation may impact soil GHG emissions associated with increases in temperature and length of the period under fire whereas non-burned areas are documented to have lower CO_2_ and N_2_O emissions compared to burned areas. The decrease in root respiration is due to differences in the related pH, and burned residue [20]. Besides these environmental factors, beneficial management practices are vital components for improved strategies to cope with the CF phenomena.

## 3. Land-Use Changes and Carbon Footprint

Soil organic carbon (SOC) is a result of C inputs and losses mechanisms whereas naturally accumulated SOC largely depends on the degree of vegetation cover, and the differences between C inputs and outputs. Changes in land use might have a significant impact on soil parameters and this can alter the source-sink balance of atmospheric GHG emissions. Also, plant species changes in root depth, plant root characteristics, and spatial distribution can strongly influence CF from soils. According to the Food and Agriculture Organization [21], the global land surface (149.4 mill km^2^) includes grassland and pastureland (31.5%), woodlands (27.7%), barren land (15.2%), cropland (12.6%), snow and glaciers (9.7%), water bodies, wetlands and mangroves (2.7%), and artificial surfaces (0.6%). Climate, technology, and economics also appear to impact land-use change at different spatial and temporal scales. Land use and land-use change directly or indirectly influence GHG emissions from terrestrial ecosystems to the atmosphere. Therefore, evaluating the CF of different land use and different climate regions separately is demanded to better understand the mechanisms behind CF.

Some of the important factors in these mechanisms can be named as vegetation types, age, and density due to their impacts on soil respiration [22]. The soil respiration rates under young spruce forest stands were reported to be higher than those under 10-, 15-, 31- and 47-year-old stands by Saiz, et al. [23] due to differences in fine root biomass and differences in microbial respiration associated with higher organic inputs. Increasing microbial respiration by organic inputs may also influence C sequestration potential. However, deforestation is the most common issue in terms of CF analysis and land uses. Deforestation and other land-use fluctuations to enhance the surface area for crop production contributes to climate change [13]. About 30–35% of the soil C stored in the top 7 cm layer of soil was lost in the first 30 years once forests were converted into agricultural fields [24]. When a forest is converted to agricultural land, aboveground C stock can be lost in various ways such as being taken away as a product (wood), loss due to combustion, and rapid microbial decomposition. Some soil organic matter may also be oxidized to C emission via tillage.

Similarly, wetlands were reported to result in a higher absolute CH_4_ emission rate than all other land uses [13]. Wetlands are responsible for 25% of the total anthropogenic and natural CH_4_ sink [21]. There are several properties such as soil moisture, water depth, temperature, and crop type responsible for CH_4_ emissions. Wetlands are globally drained for diverse reasons whereas about 50% of wetlands are changed to other land uses worldwide [25]. Due to fertilization, tillage, and oxidization, using dried-up peat soils in agriculture cause higher CO_2_ and N_2_O emissions. Protected C stocks under the anaerobic environments in wetlands can be reduced by aerobic respiration [26]. Rice fields can be a good example of wetlands in terms of CF.

Other examples of different land uses are grasslands and croplands. Permanent grasslands indicate 31.5% of the total global and 70% of the total agricultural land area [21] and emit above-average GHG emissions. Additionally, croplands have a strong influence on CF. Agriculture activities which directly or indirectly impact GHG emissions as well as C sequestration, represents 12.6% of the global land [27]. Furthermore, the global C sequestration potential of agriculture indicates 0.73–0.87 Pg C yr^−1^ [28]. The balance between C sequestration and GHG emissions shows the importance of agricultural practices. In other words, agricultural practices such as tillage and fertilization must be considered when CF or C sequestration is calculated. The higher root mass because of higher atmospheric CO_2_ content can uplift CO_2_ concentrations in soils [29]. Furthermore, higher soil moisture might be due to a decrease in the opening time of stomata under elevated CO_2_ concentration in the atmosphere, which enhances the conditions for N_2_O and CH_4_ emissions [20] and drive denitrification [30]. Soil temperatures might be lower owing to enhancements in the leaf area and an associated shade [20]. This indicates the importance of considering landscape positions, plant residues, crop quality, and hence photosynthesis capacity of those land for CF calculations.

Crop residues are used to protect soil, lower erosion, maintain soil humidity, increase soil quality, and hence impact soil emission rates [13]. One of the important components in CF under the land-use perspective is landscapes. Soil properties and hence CF from soils differentiate from summit to floodplain. Soil water infiltration decreases from the summit to shoulder following with backslope whereas infiltration starts increasing again from toe slope to floodplain. Soil erosion and insolation have opposite trends of soil water infiltration under different landscapes. Furthermore, sedimentation increases by moving from summit to floodplain. This shows how soil quality and agricultural vegetation change through landscapes. However, CF can be influenced by land uses, landscape positions, as well as soil management practices. These factors and their impacts on CF are better explained in the following sections.

## 4. Agriculture and Carbon Footprint

One of the largest sources of GHG emissions is agriculture, which emitted about 10–12% of the total global GHG in 2005 whereas this value has increased to 13.5% (CO_2_: 25%; CH_4_: 50%; and N_2_O: 70%) by 2009 (Second largest source) and to 18% by 2011 [31]. Both scientific and public importance of the CF of agricultural inventions bounds up with the quantity of GHG emissions due to agricultural management practices such as tillage, inorganic fertilization, and harvesting [32], pesticides, manuring, waste management, composting, biochar addition, and crop photosynthesis capacity. Therefore, sustainable agricultural practices need to be investigated to challenge these issues.

Controlling agricultural management by assessing the agricultural CF might be an option for mitigating GHG emissions and hence climate change. Recently, some experiments have addressed the agricultural CF under different managements such as tillage, cropping systems [33], and nitrogen fertilizer rates [10]. However, reports do not contain sufficient information concerning responses of CF of crop production to integrated technologies with different agricultural strategies.

Agricultural practices need a significant amount of energy due to machinery processes. Therefore, enhancing energy use efficiency and lowering CF related to crop production is an essential requirement. Since GHG emissions are from soils and originated from biological activities which are sensitive to soil properties [34], it draws progressively more attention to increasing production efficiency and decreasing CFs together. Many studies have documented the importance of soils to decrease conventional energy use, provide clean energy, and hence understand low-C agriculture. Reducing GHG emissions should be in place by the time the sustainability of soil health/quality is secured or improved. Due to critical direct and indirect effective components in agricultural GHG emissions, understanding the mechanisms in the complex and dynamic soil system, and their intercorrelation with climate change issues, is crucial. A better understanding of climate change impacts on SOC needs a determination of the expected influences that climate change has on the comparative extent of soil C inputs and losses. 

## 5. Role of Soil in Carbon Footprint and Agriculture

The soil is an important source and sinks of atmospheric C due to agricultural applications and management strategies on a global scale. A large amount of organic C and nitrogen are stored by soils which vary through the soil profile and cause GHG emissions associated with decomposition and microbial activities. If all other factors in the C cycle stayed steady, a difference of 1% soil C content would result in about 8 ppm CO_2_ alteration in the atmosphere, and this 8 ppm CO_2_ response might be lowered by considering the potential mediating responses due to photosynthesis and oceanic exchange [34]. It is a consequent result that global soils and soil management have the potential to either enhance or reduce atmospheric GHGs and climate change. Therefore, the relations between photosynthesis and GHG emissions as impacted by agricultural management practices especially focusing on soil and the related systems should be considered.

### 5.1. Soil Types

Soil type is a significant factor that impacts GHG emissions directly or indirectly by influencing soil structure and soil wetness [35]. For instance, N_2_O emission is reported to be higher from clay loam soils in comparison to those from loam soils [7]. Soil bulk density and clay content are significant factors for the comparison in terms of N_2_O in spring. Further, [36] reported higher N_2_O and CH_4_ emissions from Histosols in comparison to Gleysols and Plaggic Anthrosols whereas differences in CO_2_ emissions were not significant between Histosols and Gleysols. Similarly, Butnan, et al. [37] reported that total CO_2_ and CH_4_ emissions had a positive correlation with the addition of higher volatile matter that contained biochar in the coarse-textured low-buffer Ultisol but it was not correlated in the fine-textured high-buffer Oxisols. N_2_O emission had positive influences on the Mn-rich Oxisols potentially due to differences in mechanisms indicating microbial activities, soil aluminum and manganese toxicities, and soil pH impact on these soils [37]. Therefore, soil type is an important component in CF predictions besides soil health indicators.

### 5.2. Soil Health (Feedback Mechanism)

The understanding of mechanisms behind complex and dynamic soil systems is important to better understand the impacts of agricultural management practices on soil and environmental health. For instance, the application of manure as a soil amendment can be an option for enhancing soil quality and mitigating climate change [10]. However, it is more important to know how manure impacts a particular soil property and what differentiations are caused by these changes.

Even though GHG production is mainly a biological process, soil physical properties also impact biology by changing the physical environment of soil microbes. Both static and dynamic properties of soils are impacted by C which in turn affects C sequestration potential indirectly. The higher SOC and the lower soil bulk density indicate a higher degree of compaction in the soil. SOC is also significant and positively correlated with soil aggregate stability, soil structure, and erosion refers that SOC clarifies a significant amount of the variability of stable aggregates which is vital due to its positive influences on plant growth and the environment. Improving soil health indicators may result in more GHG emissions but they will also increase the C sequestration capacity of the soil through photosynthesis and hence mitigate climate change issues. Previous studies reported the significance of aggregate size distribution, inter-aggregate porosity, and gas diffusivity as leading to the degree of anaerobiosis and denitrification in soil.

Soil structure is one of the most important components in CF phenomena due to its influence on GHG emissions through microbial activity, WFPS, soil metric potential, volumetric water content, aeration, relative gas diffusivities, and air permeabilities, and restricted aeration [35]. For instance, poor structure lowers the relative gas diffusivities, and air permeabilities, and restricts aeration which are relevant indicators for N_2_O and CH_4_ flux and aeration status [35]. Moreover, Ozlu and Kumar [10] reported the relationship of volumetric moisture content of the soil with air temperature, WFPS, and hence GHG emissions. 

Furthermore, an explanation for soil temperature influences on higher N_2_O emission might be possible by anaerobic microsites as higher respiration and oxygen requirements [16]. The capacity of oxygen (C), CO_2_, N_2,_ and N_2_O (D) to exchange on soils due to pore sizes (A) and total porosity is a controlling factor for GHG emissions, Figure 1. Such mechanisms are significantly determined between aggregates partially by gas diffusion rates [35]. Arah, et al. [38] conducted research in southeast Scotland to estimate soil N_2_O emissions by using Fick’s Law and evaluated gas diffusion rates and N_2_O fluxes. In addition, crop roots are significantly important especially due to their roles in soil aggregation. Crop roots keep soil particles together and help to develop soil aggregates (B). Soil structural quality may not always be described by considering all these properties and mechanisms but it is determined from certain results of soil physical properties such as porosity, water retention, air permeability, hydraulic conductivity, gas diffusivity, aggregate stability, and penetration resistance [35]. Considering these processes and properties in soil structure phenomena may help to better understand soil structural development and its role in CF. For example, soil water retention indicates the interaction of soil moisture content and soil water potential which also influence soil redox conditions whereas soil water retention under field capacity is significantly and positively linked with SOC. Similarly, SOC is associated with total porosity, saturated hydraulic conductivity, and soil bulk density [39]. Further, increases in soil bulk density and soil strength are the results of soil compaction which also reduces soil macro-porosity and water infiltration [40].

Soil texture is also a key identifier of soil property not only due to its effects on soil structure but also on soil functions under different land use and soil management practices such as tillage and compaction. NO emissions were reported to be the highest in coarser soil textures whereas soils with finer pores cause the higher formation of CO_2_, N_2_O, and CH_4_ (under anaerobic conditions) [41]. Soil texture further provides structural hot spots for microbial activities but there is no certain proof for the correlation between aggregate sizes and N_2_O fluxes [35]. In contrast, soil aggregate stability with the soil matrix, compaction, and distribution of organic C fractions are important for C stabilization [42] and hence GHG emissions. Microbial activities, root respiration, chemical deterioration treats, and fungi activity cause higher soil GHGs [43] depending on soil pH and C/N ratio [13]. Soil pH increases have been reported to increase CO_2_ and N_2_O which are significantly impacted by management and fertilization such as liming and manure. Similarly, N_2_O emissions are negatively associated with the C/N-ratio, where CO_2_ and CH_4_ emissions are positively related to the C/N-ratio [44].

Therefore, it can be stated that the addition of N sources may increase soil water retention and compatibility owing to increases in biomass production and C input [45] but may also increase GHG emissions [33] due to higher microbial activities. Now the question is how the increase in soil health will help to mitigate the climate change issues if improving soil organic matter and soil structure itself will produce higher GHG emissions. For instance, the addition of organic amendments such as manure can improve soil health and mitigate climate change issues [46,47] by providing higher C sequestration capacity than causing increases in GHG emissions. Photosynthesis is one of the most important key processes in which C stabilization is a key property.

### 5.3. Carbon Stabilization and Storage

The SOC might be stabilized by three mechanisms in soil: physical protection, chemical composition, and biological stabilization [42]. Therefore, it is important to understand the C cycle especially before C turns into kinetic form and causes much bigger issues for our planet. Soil C storage is the largest sink of C on the planet with 2500 Pg (petagram, 1 Pg = 1015 g) C in top 1 m soil depth [48]. The C loss and soil C sequestration are two components of building this C in soils depending on management practices such as reduced tillage, good quality of pasture, green manures, manures, composts, and other sources of organic matter. The quality and quantity of soil organic matter, therefore, have a critical role in C balance worldwide.

Baldock, Wheeler, McKenzie and McBrateny [34] stated two types of the biologically stabilized SOC which are responsible for the biological capability of a particular form of SOC and the indicate biological capacity (decomposition) of SOC depending on biochemical recalcitrance, functional capacity, genetic potential, environmental properties, biochemical reactions, and physical protection of soil. In addition, the most labile organic matter fractions are water-extractable organic materials which are critical sources and influence CO_2_ emissions. Stable organic matters in soils, which are resistant to decomposition and stay in soils for a long time, may be referred to as humus. Owing to their roles in soil physical protection, aggregate formation and cation exchange capacity are vital to stabilizing soil organic matter [39,42]. Due to their role in decomposition, higher microbial communities generated humus over time increasing soil health which provides improvements in healthy crops, yields, profits, stable and good soil structure, and thus C sequestration [39]. Therefore, healthy soils do not necessarily decrease GHG emissions but increase C sequestration more.

## 6. How Does the System Work?

Recently many research experiments and greenhouse studies have been conducted to determine the energy use and CF of different agricultural practices including crop rotations, tillage practices, manure application, integrated crop-livestock systems, and cropping systems such as the rice-fallow [49], open field tomato (Solanum *Lycopersicum*) production [50], cotton (*Gossypium* spp.) production [51], conventional and organic farming systems [52] and plant and animal-based food products [53] worldwide.

Challenging climate change, CF relies on inputs and outputs through the soil system. Qi, Yang, Xue, Liu, Du, Hao and Cui [32] also highlighted that only input, output, and unit developments ought to be involved in CF evaluations due to their direct association with the product. Thus, to determine the CF and mechanisms responsible for CF under different farming practices to advance sustainable technologies and challenge climate change impacts is respectable. The optimization of agricultural management strategies such as planting, tillage, crop diversity, and source and amount of fertilization may provide an option to mitigate GHG emissions in agricultural lands. Therefore, CF and its feedback on agricultural management strategies are necessary to determine strategies for climate change issues by both realistic and mechanistic methods. This view can be categorized into two different pathways as inputs and outputs.

### 6.1. Inputs (Carbon Sequestration)

The inputs-outputs prospectus can be understood by evaluating the mechanisms of C sequestration and stabilization under increasing temperatures and environmental conditions. Adoption of the best management practices (conservation tillage, fertilization, bio-solids or organic amendments additions, manuring, crop rotation, and improved residue management) can provide higher C sequestration in agricultural fields. The crop density, crop type, and hence photosynthesis become important aspects in CF evaluations by considering mechanistic processes which lead to the natural sequestration of soil C. Accumulation of C in soil by naturally sequestered processes might be more stable than conventional practices but further studies are necessary.

For instance, (Figure 2) explains how the soil system works under C sequestration phenomena by considering the input-output approach. In Figure 2, green-colored processes (photosynthesis, residue decomposition, biological transformation associated with N cycling, assimilation, immobilization, and metamorphic organisms) are responsible for C sequestration. Photosynthesis is the most important process in terms of C sequestration and hence decreases CF.

#### 6.1.1. Fertilizers

Before going into other options, a better understanding of how inorganic and organic fertilizers are mechanistically associated with soil structure and GHG emissions (Figure 3) is critical. Thus, realistic policies and recommended rates of fertilizer applications should be found and admitted to advancing sustainable agricultural strategies. It is well documented that balanced fertilization with chemical fertilizers and manure together increases soil nutrients, crop yields, crop growth biomass, and hence SOC contents. However, inorganic fertilization alone may not always show the same results.

**Figure 2 biology-11-01453-f002:**
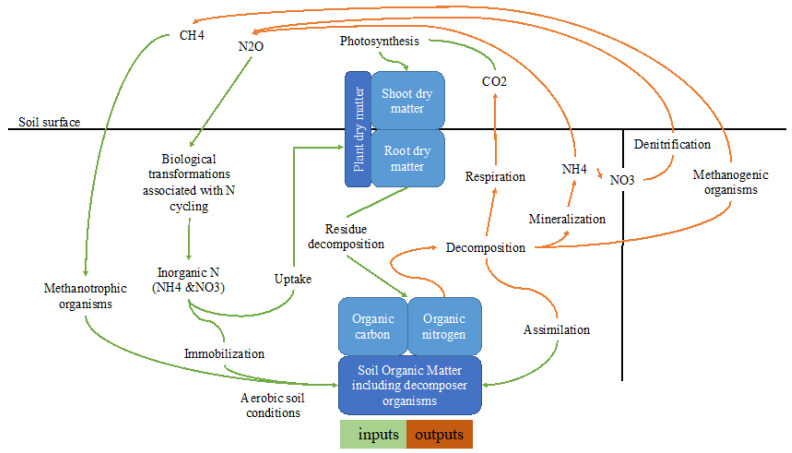
Soil biological processes that influence (inputs) utilizing of atmospheric gases into the soil and (outputs) GHG emissions from the soil into the atmosphere [34].

It is known that inorganic fertilizers decrease soil pH. Qi, Yang, Xue, Liu, Du, Hao and Cui [32] stated that acidification and eutrophication are environmental hotspots for maize production. Similarly, Haynes and Naidu [54] studied the impacts of chemical fertilizer additions and reported negative influences on soils such as involving Na^+^ which causes the dispersion of soil colloids, declines soil pH and soil moisture and enhances accumulated NH4^+^ concentrations. Moreover, chemical fertilizer applications do not significantly impact soil bulk density but also do not provide better soil aggregate stability and structure [55]. Chemical fertilizer applications add available nutrients to the soil but negative influences might appear due to its exclusive properties [56]. Due to Al or Ca phosphate-binding formation, phosphoric fertilizers and phosphoric acids may positively influence aggregation but, more importantly, NH4^+^ addition at high concentrations may cause a dispersion by moving into soil clay colloids [54]. As explained above, too much inorganic fertilization causes soil degradation and a decline in soil structure. This will increase GHG emissions but its positive impacts on C sequestration are questionable.

As shown in Figure 3, chemical fertilization is associated with increased crop yield, decaying organic materials, and binding soil particles. However, inorganic fertilizer also contributes to GHG emissions. The C sequestration/GHG emission ratio is important in terms of CF phenomena, especially under chemical fertilization. In addition, the higher cost of chemical fertilization and the degradation of native soil fertility are indicating greater interest in low-cost organic fertilization such as manure for C and soil nutrition [57]. This makes inorganic fertilization alone less attractive compared to organic amendments.

#### 6.1.2. Manure Applications

Manure addition is a more successful approach than inorganic fertilizers to increase the SOC stocks. The rates and types of bedding material, time of buildup, water quantity and quality, location, and length in storage can influence the quality of manure [58] and can cause a wide range of nutrient contents [59]. For instance, Ozlu and Kumar [60] and Ozlu, Kumar and Arriaga [47] reported that not only the SOC at 0- to 40-cm soil depth was increased by overall manure addition in comparison to chemical fertilizer and control but also increased rates of the manure enhanced SOC content which was positively associated with total nitrogen, electrical conductivity, and water-stable aggregates in two eight and twelve years’ experiments. Similarly, [61] reported that stable C content as impacted by manure was significantly greater compared to inorganic fertilizers whereas manure additions provide higher C sequestration in fine soil fractions over 17 years. Therefore, the addition of manure provides higher soil C sequestration, and better soil health but not inorganic fertilizers.

Manure has been confirmed as a positive contributor to soil health due to its influences on soil properties such as biological indicators, soil microbial community compositions, microbial biomass, earthworm populations, enzyme activities, [62], and soil physical properties, soil aggregation, porosity, soil bulk density, and compaction, maintaining soil pH and improving soil water relations [46]. For instance, a decrease in soil organic matter also decreases water-holding capacity, aggregation, soil structural stability, porosity, and decomposition rate but increases soil compaction and affects erosion. Manure, therefore, enhances soil organic matter which is a significant factor in crop production and hence helps to mitigate climate change and create a win-win scenario. As seen in Figure 2, manure addition increases SOC directly or indirectly by providing better soil structure. All mechanisms and benefits of manure additions above explained somehow contribute to C sequestration and challenge the CF of agricultural fields.

#### 6.1.3. Biochar

Biochar (pyrolyzed biomass) application on soils is an approach for C sequestration to stabilize the C which is adapted by plants [63]. Biochar can also sequester C and increase soil and environmental quality, soil fertility, soil structure, nutrient availability, soil-water retention capacity, and hence C storage capacity [63]. Moreover, biochar is negatively charged due to their structure (containing phenolic and carboxyl groups) and enhances surface negative charge and cation exchange capacity [64]. Therefore, plant-derived C (composts or biochar) applications may sequester C in soils and remove CO_2_ from the atmosphere together. The addition of C through biochar applications can accumulate in soils for several decades or longer, even though stability might depend on several factors including the amendment type, nutrient content, and soil physical structure. Therefore, the addition of biochar is recommended to reduce soil degradation and soil-borne GHG emissions, increase C sequestration and soil nutrient contents, and hence challenge climate change issues, [65].

#### 6.1.4. Crop Residues

Crop residues accumulate in the soil system from the top layers to the lower depths by deposition (shoot residues) or through root residues, exudates, and root-associated mycorrhizal fungi [34]. SOC content enhances by the decomposition of organic substances such as crop residues at the surface layer of the soil profile and hence improves soil aggregation. Besides chemical fractionations of SOC, particulate organic C (organic C associated with particles >50 mm), humus organic C (organic C associated with particles <50 mm), and resistant organic C (organic C found in the <2mm soil and having a poly-aromatic chemical structure) have been used when referring to allocate SOC interactions to soil physical and chemical properties Skjemstad, et al. [66]. The C sequestration needs crop residue/biosolids additions or fertilizers/manures applications to increase crop growth biomass production. These studies showed that replacing the stabilized C with easily decomposable C content is important to improve sustainable soil health and reduce the CF in agricultural practices.

#### 6.1.5. Photosynthesis

Photosynthesis (C dioxide; 6CO_2_ + water; 6H_2_O → sugar; C_6_H_12_O_6_ + oxygen; 6O_2_) is a process that runs by chloroplast in crops and produces a significant amount of oxygen and sequesters C in sugar form and transferred to soil either by roots or decomposition of plant residues. Sugars as a result of the photosynthesis process can help nutrient uptake through sugar sensing, normalize numerous root functions, and be transferred through roots (indicating glucose and sucrose pathways) [67] into soil organic and inorganic C pools and hence be long-term sequestered. Roots help C sequestration depending on several factors including; (i) rising light interception efficiency, (ii) enhancing solar energy adaptation to biomass, (iii) higher C portion to roots, (iv) tolerance to biotic and abiotic stress, (v) promoting biomass quality, and (vi) growing high-yielding perennials for agriculture [68]. Transferred C through photosynthesis increases organic matter content and hence enhances soil fertility, improves structure, provides healthier crop production, and increases the C sequestration potential of those soils. By advancing farming practices with careful C-friendly management strategies, photosynthesis can be enhanced to a level at which maximum C sequestration takes place.

Environmental factors affecting photosynthesis are light, CO_2_, temperature, wind, H_2_O, and nutrients. Under the above conditions, 50% of CO_2_ emissions since 1750 have been recycled into the oceans and terrestrial ecosystems [69]. Significantly high CO_2_ concentrations in the atmosphere enhance the photosynthesis rate, rubisco activity, carbohydrates, and biomass production, and hence crop productivity depends on the accessible and transferable soil nutrients in the rhizosphere [70]. Increases in carbohydrates enhance crop growth, starch reserves, auxin biosynthesis, stimulation of gene transcription, and finally root growth which also depends on interactions of carbohydrates with hormones [71].

Soil moisture content is one of the impacts on the variation in photosynthesis and biochemical factors more than climatic conditions. Soil biochemical such as mycorrhizal fungi also help crop roots to penetrate deeper into soils and send out their below-ground networks (hyphae) to generate efficient secondary root systems to allow the plant to access moisture [72] whereas glomalin (glycoprotein) can improve the agglomeration of soil particles, and hence increase water retention. This creates a symbiotic relationship between fungi and crops (photosynthesis) to produce and deliver the sugars for roots to grow [72]. However, dry conditions such as water stress, continuously influence crop metabolism and lower crop growth and photosynthesis [73]. Water stress lower photosynthetic assimilation of CO_2_ due to limited CO_2_ diffusion in the leaves, stomatal closure, inhibition of CO_2_ metabolism, and the amounts of ATP [73]. Moreover, lower photosynthetic assimilation of CO_2_ is also caused due to inhibition of the Calvin (photosynthetic C reduction) cycle, but it is still demanded what biochemical processes are most delicate to stress conditions [73]. In this case, PGPR (Plant Growth-Promoting Bacteria) can help to improve the abundance of nitrogen and soil nutrients in the crops’ root systems, decrease water consumption, and enhance metabolic functions [72]. Furthermore, bioenergy crops can be an option to mitigate GHG emissions and sequester C into the soil system via their wide root systems and their properties such as photo-assimilation of CO_2_ [68].

### 6.2. Outputs (Carbon Emissions)

The inputs-outputs prospectus can also be an option to better understand or evaluate mechanisms responsible or sources for each soil CO_2_, N_2_O, and CH_4_ emissions and estimation of their impacts on climate change. Then, by considering C sequestration, important aspects of CF may be better understood. For instance, the above figure (Figure 2) also explains how soil biological processes produce GHG emissions. Following the decomposition of organic matter, the respiration, mineralization of C, denitrification, and methanogenic organisms and their activities are the pathways through which those soils lose the C to the atmosphere.

#### 6.2.1. CO_2_ Emission (Soil Respiration) 

The CO_2_ emission is the major contributor to soil GHG emissions produced from active organic matter which naturally gathers C from the addition of crop residues, roots, and exudates, and decomposes by soil micro-organisms [74]. Due to its role in losing SOC under agricultural practices and contributing to CF, the quantification of these CO_2_ emissions is important. Even though net quantification of CO_2_ fluxes can be determined, high technology equipment and data analysis are necessary which do not measure the contribution of individual processes [34] such as microbial activities and the mixture of decomposition and heterotrophic respiration [75]. CO_2_ fluxes can be originated from two mechanisms: (i) root, anaerobic, and aerobic microbial respiration, and (ii) aboveground plant respiration [13].

The CO_2_ emission increases with higher temperature and maximizes in the summer whereas it decreases with lower temperature and is lowest in the snow-covered winter [10] which indicates higher CFs during the growing season. Besides temperature, soil moisture was reported as an important factor in CO_2_ emission. CO_2_ emission had been documented to decrease after heavy rainfall possibly due to poor gas diffusivity and air-filled porosity under an increased anaerobic environment [76]. However, differences in WFPS did not show any significant influence on CO_2_ emissions in the sandy loam or clay soils over 60 days of incubation [77], and in some other field experiments. Finally, Ball [35] and Ruser, Flessa, Russow, Schmidt, Buegger and Munch [77] reported no impacts of soil moisture content on soil CO_2_ emissions under different compaction levels, except those soils near to saturation (>98% WFPS). In addition to this statement, Ball [35] stated that CO_2_ emissions were the highest on the best structured and sandy loam soils apparently due to the loose, well-aggregated structure which provides good aeration. Therefore, physical protection of soil aggregates, environmental properties, and relatively soil clay content might be what makes soil structure important in CO_2_ emissions. Due to the role and amount of the soil clay particles soil texture come into the mechanisms and influence CO_2_ emissions. The clay soils were reported to have three times’ higher respiration which indicates diverse nature and more decomposable soil organic matter fractions in these soils [78]. Similarly, [79] found low CO_2_ emissions from sandy and clayey soils. These changes in CO_2_ emissions related to clay content might be due to the chemical composition of soil which refers to soil pH and cation exchange capacity (CEC). The soil pH and CEC were found to have empirical relationships with CO_2_ and N_2_O production depending on soil moisture [78].

Moreover, agricultural practices influence CO_2_ emissions both directly and indirectly by changing soil health indicators and hence soil structure. For instance, organic and inorganic fertilization are major sources of GHG emissions in agricultural fields. To decrease GHG emissions, reducing the inorganic fertilization rates and enhancing the fertilizer use efficiency might be an option (Qi et al. 2018). Similarly, application sources (organic or inorganic) are important for CF. For instance, Ozlu and Kumar [10] reported that manure may lower N_2_O emissions more than inorganic fertilizer applications. With higher yield and lower CF, manure represents a possible C-friendly agricultural management practice that balances the environmental burden and crop production. To place this goal, minimizing GHG emissions and maximizing crop yield and soil health together is a clear objective. However, intensive chemical fertilizer additions might be overpriced, increase nitrate pollution, and decrease SOC content [46]. Therefore, there is a strong necessity for alternative fertilization strategies at the recommended rate to avoid negative influences on the soil structure and the environment (Figure 3). Furthermore, Severin, Fuß, Well, Garlipp and Van den Weghe [36] reported an increase in CO_2_ emissions during the first 24 h to 48 h after chemical fertilization when fatty acids from slurries are metabolized. In addition, Ozlu and Kumar [10] documented a high rate of CO_2_ emission under chemical fertilizer application which continued for about 20–25 days whereas those under manure additions increases slowly and continued to about the growing season. Similarly, high CO_2_ emissions during the first days after the addition of manure might be due to bacterial soil organic nitrogen mineralization and denitrification [80]. Besides fertility practices, this may also vary due to soil types. For example, Histosols tended to have higher CO_2_ emissions owing to their high ability of mineralization and denitrification with high SOC and water contents [36]. Organic amendments such as manure are among the sources contributing to significant quantities of GHG emissions. Ozlu and Kumar [10] reported significant influences of not only overall manure and inorganic fertilizer but also increased rates of manure additions on the annual CO_2_ and N_2_O fluxes but not CH_4_ in 2015 and 2016. Manure has complex organic compounds which are decomposed by bacteria and produce CO_2_ emissions in aerobic environments [10].

Similar to fertilizer use efficiency and using organic sources besides chemical fertilization, integrated cropping systems might be a good option to challenge high CF and hence climate change issues. Innovations such as annual pastures with rotational grazing, and the adoption of pasture–cropping systems might increase soil C content in agricultural fields and help to mitigate climate change issues. Cover crops and crop residue strategies in these, integrated crop-livestock, systems can significantly help. This C return to the atmosphere in the form of CO_2_ depends on its SOM recalcitrant property, integration with decomposer tissues, and interactions with soil minerals [34]. It is known that crop residues are a stable form of C due to the duration of decomposition.

Another stable form of C is produced because of the pyrolysis process called biochar which is used in agricultural fields both individually and combined with other sources such as manure. Biochar’s influence on GHG emissions depends on several factors such as biochar properties, biochar rates, soil texture and mineralogy, their interactions, volatile matter, and ash contents. Biochar reduces N_2_O emissions but enhances CO_2_ and CH_4_ emissions due to high-volatile matter, which has toxic influences on nitrifying and denitrifying microorganisms [81]. The CF of biochar is very low (less than 15%) with more than 95% of optimistic influences such as the cultivation phase, pyrolysis, palletization, and packaging process [63].

Organic amendments such as manure and biochar were reported to have positive influences on soil pH, relatively on crop growth, and hence CO_2_ emissions. Higher soil pH might inhibit microbial CO_2_ and denitrification processes whereas N_2_O fluxes can be increased if pH is higher than neutral [82]. Further, higher soil pH decreases the methanogenic activity [83] and hence influences CH_4_ emissions. The CO_2_ emissions under biochar addition were reported as being produced by the rapid decomposition of labile organic complexes by co-metabolism with microbial enzymes [84], biochar acting as foci [85], increasing soil fertility advancing microbial growth [86], functioning organic molecules as an oxidizing agent [87], declining extractable NO_3_^−^ (Figure 2) or immobilization to support microbial activities and decomposition [88]. Similarly, CH_4_ emissions were enhanced by ethylene on methanotrophic bacteria due to their inhibitory impact [81]. Biochar may impact CH_4_ emissions by adding volatile matter by (i) CO_2_ and acetic acid production due to aerobic decomposition of SOM mixtures or (ii) inhibiting methanotrophs [89]. Some anaerobic microbes can even be active under low soil moisture and hence produce CH_4_ [90].

#### 6.2.2. CH_4_ Emission

Methanogenesis plays a critical role in the biogeochemical cycle of C by contributing to CO_2_ and CH_4_ emissions [3] and hence the CF of GHG producing environments [74]. CH_4_ is oxidized in soils under aerobic conditions by methanogenesis, [91] which is usually most active at 1-m soil depth [35]. There is a significant relationship between CH_4_ and CO_2_ owing to similar sources or mechanisms, such as enteric fermentation and ruminant respiration [46]. Soil microbial activities enhance under aerobic circumstances and decrease with the lower oxygen availability [92]. It is obvious that soil physical properties play a critical role in CH_4_ uptake associated with CH_4_ oxidation rate, air permeability, and gas diffusivity [91]. However, these associations may not always be significant under particular conditions due to higher influences from other soil properties such as soil pH, moisture, temperature, nitrogen, and organic matter type and content [35].

When redox potentials are lower than –100mV, a significant amount of CH_4_ can be released from soils, depending on the level of saturation [34]. In this case, soil properties are more effective when they control the rate of oxygen diffusion (e.g. soil bulk density and pore size distribution) and consumption (e.g. presence of decomposable C substrates) [93]. Therefore, as representing the property for soil structure, soil moisture content and WFPS utilizes a solid influence. Dalal, et al. [94] stated that increased methane consumption rates range between 80 to 30% WFPS. Similarly, temperature increases also result in higher CH_4_ production depending on environmental factors and biologically available substrates [94]. Thus, irrigated agricultural production with increases in temperature during the growing season becomes a potential source of CH_4_ production whereas other water management strategies such as non-flood irrigation (e.g. drip irrigation) play a vital role to minimize CH_4_ emissions from agricultural fields [34].

Chemical fertilizer additions do not significantly influence CH_4_ emissions [95]. This might be due to very low rates of CH_4_ production from soils that have high aeration in agricultural fields. Similarly, soils under inorganic fertilizer applications (high ammonium concentration) contain less methanotrophic bacteria whereas manure addition increases the population of methanotrophic bacteria [62]. As the largest C source of GHG emissions produced from manure, the CH_4_ is influenced by the rate of manure applications and the portion of the manure that decomposes anaerobically [10]. In addition, the major contributor to CH_4_ production is livestock. Therefore, integrated crop-livestock systems need further focus on CH_4_ emissions. Cover crops and crop residue management come up to attention due to their important roles in both agriculture and livestock. Crop residues with a low C:N ratio increase N_2_O emissions under aerobic conditions but this may not be observed in the anaerobic environment [38]. 

#### 6.2.3. N_2_O Emission

The N_2_O emission, which is a pertinent GHG even at low concentrations since its high global warming potential (298 times of CO_2_) than CH_4_ and CO_2_ emissions, causes stratospheric ozone depletion. Soil N_2_O emissions are a result of two natural and biological mechanisms including the transformation of inorganic nitrogen (nitrification), and denitrification (conversion of nitrate to N_2_O and N2 gases) [96] associated with soil moisture, temperature, pH, SOC and N contents, texture, mineral N, microbial activities, aggregation, and structure of aggregates.

The N mineralization which provides plant-available nitrogen is also associated with some environmental factors and temporally with crop requirements (Baldock, et al., 2012). Soil N_2_O emissions increases with an increase in soil temperature [10] which removes the limitations in soil moisture content especially under irrigated systems unless soil inorganic nitrogen contents are firmly regulated and hence nitrification and denitrification rates are decreased [34]. Moreover, Dalal, et al. [97] reported that generally comparative N_2_O emission rates were not significant at less than <40% and more than >90% WFPS, and maximized from 60 to 70% WFPS. This might be because of that nitrification and denitrification can concurrently be active under aerobic and anaerobic microenvironments in soils where WFPS ranges from 60% and 80% and denitrification increases when WFPS is higher than 80% [96]. Similarly, even though denitrification is the main source of N_2_O emissions, aerobic and anaerobic conditions can expand in some aggregates and nitrification turns out to be a considerable factor in N_2_O emissions [98]. Previously studies also showed that sources for N_2_O emissions are at the top layers of profile and N_2_O production places at 20-25 cm soil depth, therefore, the soil structure and WFPS become very important [43] especially because of its ability to allow the infiltration of added fertilizers by rainfall or irrigation under the root zone. Furthermore, this dissolved N_2_O can also be lost by (i) being reduced to N2 and uptake by crop roots hence playing role in N_2_O emission through plant transpiration [43], and (ii) entering the drainage water and quickly transferred to the atmosphere [99] depending on soil compaction and soil texture.

Some studies and field experiments [78] agreed that the total N_2_O emissions from the clay soils are generally lower than those from the sandy loam soils but some others do not [100]. The contrast to this statement might be because of that soils with high CEC (e.g. clays) may enable NH_4_^+^ immobilization [101]. In contrast, the coarser textured (sandy loam) soils may support N_2_O emissions due to increased nitrification [102]. Finally, Ref. [35] stated that soil N_2_O emissions have been associated with soil structure by working on the silty clay, and the sandy loam soils which indicate that soils like silty clay have a higher potential to emit N_2_O than large sand soils. Similarly, soil type is an influential factor in N_2_O emissions, for instance, soils like Histosol have a higher potential for N_2_O emissions compared to the others, Gleysol and the Plaggic Anthrosol [36]. This might be because of easily degradable organic C, inorganic N, and lower gas diffusivity but higher soil moisture content and microbial respiration of the Histosols [103].

Furthermore, agricultural management practices might also be the reason for differences in N_2_O emissions. Even though higher soil pH, temperature, moisture, SOC concentration, and oxygen supply individually provide higher GHG emissions, combined impacts of soil moisture, SOC concentration, and microbial inhabitants on N_2_O emissions are not readily predictable. Chemical fertilizers influence soil N_2_O emissions by changing microbial decomposition and root respiration due to nitrification and denitrification [104]. Chemical fertilizer had been well documented as increasing N_2_O emission not only by overall influences but also because increased by increasing rates of inorganic fertilizer increase the N_2_O and CO_2_ emissions [10]. Bhatia, et al. [105] documented a 28% higher global warming potential due to chemical fertilizers under the rice-wheat cropping system in the Indo-Gangetic plains, mostly dominated by N_2_O emissions. As a result of these processes, inorganic fertilizers provide higher CF due to their impacts on soil health.

Similarly, if liquid organic fertilizers are applied to the soil surface about 20–40% of the total ammoniacal N might be lost [106] whereas injection decreases the loss of NH_3_ by about 2% of total ammoniacal N owing to increases in denitrification by creating anaerobic zones [107]. Therefore, not only what sources in what quantity of organic or inorganic amendments are effective but also what technique to apply these fertilizers in agriculture is an important component to sustain soil and environmental health, and crop production together. Agricultural management strategies should be well understood before being applied in the field. For instance, soil CO_2_ and CH_4_ have different mechanisms and sources in comparison to N_2_O emissions. Soil CO_2_ and CH_4_ fluxes under manure applications are due to organic matter degradation but N_2_O is largely due to nitrification-denitrification, NH_3_ volatilization, nitrate leaching, and afterward transformed to N_2_O [108]. In general, manure, additions enhance SOC, soil NH_4_^+^ content, and crop yield but also increase denitrification rates with N_2_O losses when soil moisture is high [103]. Similarly, due to the decomposition of organic substances which generates labile C pools and increase the denitrification rate [109], manure influence N_2_O production. The liquid manure addition causes increases in N_2_O emissions in an aerobic environment, however, NH_4_^+^ fertilizers produce higher N_2_O fluxes in the saturated environment [110]. Therefore, to minimize N_2_O emissions from agricultural fields, fertilizer additions need to be applied under consideration of crop nutrient requirements [46] to challenge N_2_ emissions owing to the fact that not all forms of N can be uptaken by crops [13]. 

#### 6.2.4. Carbon Leaching

Dissolved organic carbon (DOC) is one of the critical components of the C cycle [111] which plays a significant role in the uptake and loss of CO_2_ in terrestrial ecosystems and climate change [112]. Thus, predictions in CF with higher confidence intervals are fundamental to increasing the understanding of global climate change, suitable mitigation strategies [113], and soil C sequestration. In addition, associations between DOC dynamics with soil formation [114] and C sequestration especially due to DOC mobilization and transport are increasingly popular topics [115]. Isotopes studies have reported that dissolved organic matter (DOM) generally indicates organic matter addition not only due to the decomposition of recent crop residues but also owing to humified organic matter (HOM), especially from the O horizon [116]. Therefore, C leaching indicates a relatively lower amount but is significant since it is a continuous process of carbon loss from terrestrial ecosystems.

The specific surface of soil’s physical structure surfaces play a critical role in DOC transformations in soils. Soil clay content, porosity, soil water retention, degree of saturation, and WFPS may influence DOC associated with soil macro-pores and soil bulk density. A relatively stable degree of soil water saturation may influence the diffusion gradient of CO_2_ from microbial communities and/or the transformation of potential CO_2_ into the kinetic form depending on equilibrium concentrations [114]. Similarly, soil pH influences C leaching associated with soil mineralogy and high concentrations of extractable Fe or/and Al [111]. Higher soil pH increases DOC due to its influences on soil DOC solubility [117]. There is a positive correlation between soil C/N ratio and DOC [118] and CO_2_ fluxes in terrestrial ecosystems [119]. The C/N ratio in soils depends depend on microbial communities [118], organic matter quality and quantity due to the production of soluble residues, retention in B horizons, the ratio of C and total oxalate-extractable Fe+Al, and carbonate equilibrium [113]. Isotopes (14C) studies also stated the importance of SOC, and the DOC draining through the soil profile [114].

Not only does land-use influence the C leaching from terrestrial ecosystems [113], but also agricultural practices such as manure additions, cover crops and integration of crop residues [120], and biochar additions enhance DOC concentration in surface runoff water. Black C (biochar) can transport across the soil macropores and DOC leaching depending on soil texture, soil mineral fractions, soil structure, and water flux, [121]. In addition, C leaching was documented as increased by no-till + cover crop practices in comparison to conventional tillage + cover crops management [122]. However, no significant influences on DOC due to crop type, crop rotation, and types and rates of the N-fertilizer under tile-drained agroecosystems conditions were also reported [123]. Even though additions of N were reported not to have any significant influence on DOC concentration in lower than rooting zone [124], DOC is recently stated to be in decline from the addition of N from that layer [111]. Moreover, the application of lime in combination with manure did not affect the soil dynamics and DOC leaching but increased CO_2_ emissions [125]. Enhancing DOC is mainly biologically driven and associated with variation in the decomposition of organic substrates [126], and enhanced enzymatic activity [127]. In addition, C mineralization can cause higher DOC loss [128,129]. It is known that dissolved organic matter in soils is from soil humus, plant litter, microbial biomass, root exudates, urine and feces, and applications of organic fertilizers such as manure [130], and temporally and spatially controlled by several biotic and abiotic components such as soil type [131], soil properties, climate, crop types and agricultural management practices mentioned above [132].

#### 6.2.5. Pesticides and Herbicides

Using pesticides in agriculture increases GHG emissions and hence causes climate change, which indicates lower than 1% of total GHG emissions [133]. Predicted C emissions (kg CE kg^−1^) of the active ingredient of herbicides is 6.3 followed by insecticides with 5.1, and fungicides with 3.9 [134] whereas it is 0.35 kg CE kg^−1^ for post-production C cost for pesticide additions [135]. Gan, et al. [136] reported pesticides and P fertilizers contributed to GHG emissions more than crop residue decomposition whereas [137] reported that GHG emissions due to N fertilizer additions were nine times higher than those associated with pesticides and eleven times those owing to tillage practices. The use of pesticides is globally increasing day by day, especially in India, China, Brazil, and other developing markets [134].

Using herbicides and fungicides such as boscalid, bromoxynil, glyphosate, imazamox, imazethapyr, pyraclostrobin, and sethoxydim, which are routinely used in the production of field crops, is reported to enhance N_2_O emissions [136]. However, it is important to understand the complex interactions of affective components in N_2_O production associated with herbicides. The fundamental presentation of herbicides as phenoxies was in 1945, and it is still the major strategy for weed management [138]. For instance, herbicides constitute about 85% of the pesticide contribution for cereal farming in the northern Great Plains, USA [139].

Mancozeb and chlorothalonil, which had been found to be inhibitors for the nitrification process for different levels by laboratory studies on soil microcosm [140], may cause a significant amount of N_2_O emissions as fertilization, tillage, or irrigation involves its application during the growing season, [141]. However, the CF, due to pesticide use in the arid areas, is commonly low [136]. Similarly, crop diversification and agronomic management practices can be an option to considerably decrease pesticide applications, CF in agricultural fields, and hence improving crop health, although pesticides are not a major effect on CF [136]. Similarly, soil microbial components may help to increase nutrient use efficiency through PGPRs and interactions of numerous fungi and bacteria, and work as biofertilizers and biopesticides [142]. Herbicide use can also be limited by adding only at the particular times in weed growth and using with conservation tillage [143].

#### 6.2.6. Tillage

Soil physical protection through soil aggregates is one of the mechanisms of stabilizing soil C. Tillage leads to CO_2_ emission promptly because tillage breaks down the soil aggregates, uncovering labile organic matter, and hence increases the activities of soil micro-organisms to oxidize SOM [144,145]. Tilling soil may enhance the mineralization of SOC and hence the CO_2_ emission [146]. Previous studies reported that CO_2_ emissions are significantly more sensitive to the soil moisture content under conventional tillage compare to those under no-till practices [147]. A similar correlation was found for N_2_O emissions [148]. In addition, the no-tillage technique was reported to cause higher GHG emissions than conventional tillage potentially due to the decomposition of the organic residues on the surface under no-tillage and hence higher microbial and invertebrate respiration.

The CO_2_ emissions owing to soil tillage practices are highly correlated with the intensity and the volume of soil disturbance [147]. Disturbing soil by tillage practices may change the soil porosity, pore size distribution, soil thermal conductivity, and hence soil temperature. Due to the intensive traffic, conventional agricultural practices may cause structural degradation such as compaction. In most agricultural fields where conventional tillage is operated, residual compaction damage is visible due to the former tractor route both for the surface and subsurface soil depth. Under these conditions, the soil structure is very poor with minimal macroporosity and large moisture contents which results in anaerobic conditions and large N_2_O emissions [35]. The compaction in soils also reduces the crop yield, photosynthesis, and hence C sequestration whereas it increases the CF of these fields. Therefore, decreasing the tillage intensity or not tilling the soil may result in structural stability and turn helps to improve organic matter sequestration with a significant labile fraction and C stability. The SOC is physically attached to the surface of soil particles (clay and silt) and stabilized within aggregates in the form of recalcitrant C with the help of crop roots, microbial activities, glomalin production, and water [149]. 

#### 6.2.7. CO_2_ (Tractors), Harvesting, and Runoff

Another important contributor to CF phenomena from agricultural soils is on-and-off farming activities related to crop production and mostly forgotten when making CF calculations. These activities include farm machinery which causes GHG emissions directly or indirectly due to energy requirements [150]. The direct energy requirements indicate diesel or gasoline fuels, electricity, and gas, whereas indirect energy requirements are production inputs (seeds, fertilizers, feed, etc.), and manufacturing inputs (buildings, machinery, etc.) [151]. In addition, tillage is energy-intensive agricultural management that strategy indicates about 30% total energy requirement for crop production [152]. Therefore, these operations should be considered when CF is calculated or evaluated.

## 7. Carbon Footprint Calculations

Carbon footprint, the total amount of GHGs produced for a given activity, provides the opportunity for environmental efficiencies and cost reductions. The global warming potential of GHGs is expressed in terms of the impact on global warming of the equivalent weight of CO_2_-equivalent (CO_2_e). One unit of CO_2_ gas has ~1 unit of CO_2_e and one unit of CH_4_ and one unit of N_2_O have ~23 and ~298 units of CO_2_e. The global warming potential of all gases together commonly indicates the CF per unit area-kg CO_2_e ha^−1^.

## 8. Conclusions

The study demonstrated the effect of environmental factors, land use, and agricultural practices on C footprint management through a detailed review. This study highlights that for lower CF, healthy soils have many benefits for both the general public and especially the farmers, such as being stable and resilient, resistant to erosion, easily workable in cultivated systems, good habitat for soil micro-organisms, fertile and good structure, and large C sinks. Tillage is harmful to soil structure by oxidizing C and causing GHG emissions. If possible, no-till, if not, minimum tillage frequency and depth of tillage, and optimum moisture are recommended. The soil should be at appropriate moisture when tillage is placed. Production crops that are good for the soil structure may help to build C. Diverse cropping systems are better for the soil than monocultures. Minimizing machinery operations can help to avoid soil compaction. Building SOC in the most stable form is the most efficient practice of sustainable crop production.

## Figures and Tables

**Figure 1 biology-11-01453-f001:**
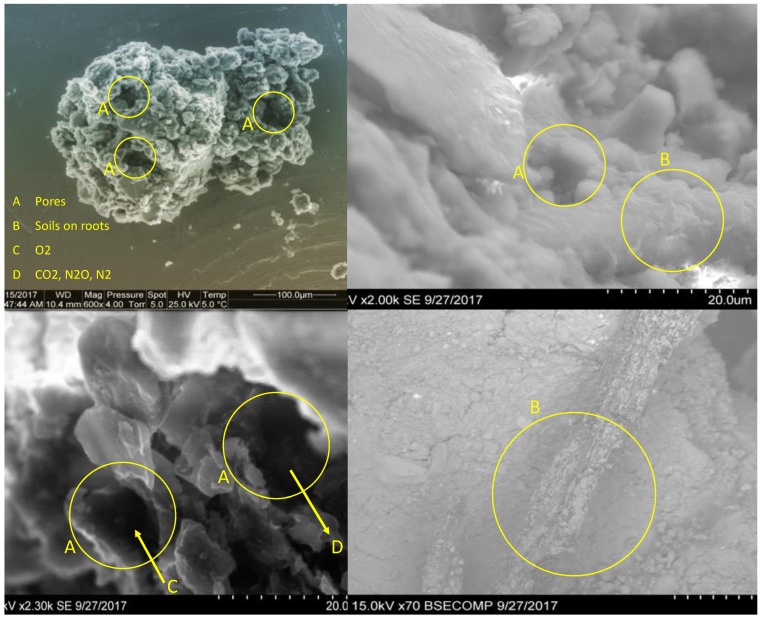
Soil aggregates and gas exchange within the soil pore system by SEM images.

**Figure 3 biology-11-01453-f003:**
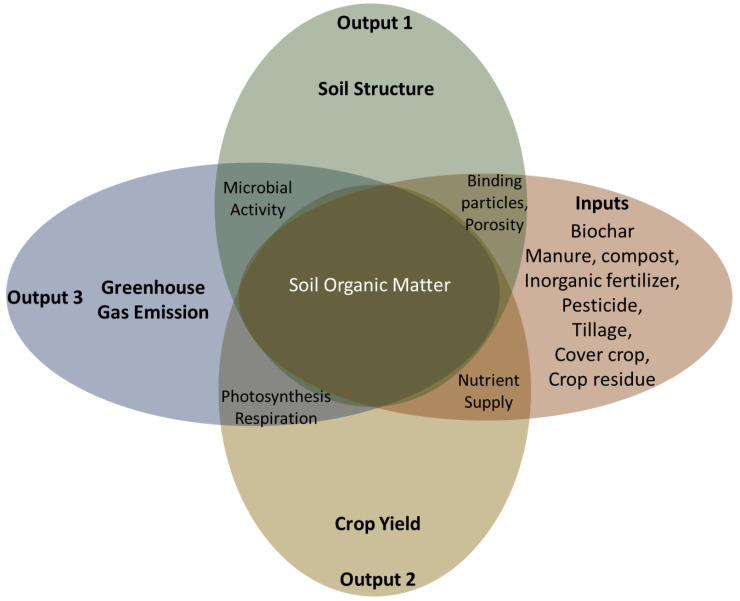
The carbon movement through soils and coupled correlation with other soil and crop properties.

## Data Availability

Not applicable.

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
