# Peer review of "Carbon Footprint Management by Agricultural Practices"

_biology, 2022, doi:10.3390/biology11101453_

Round 1
Reviewer 1 Report
Respected editor,
It is a great honor for me to review an MS entitled; “Carbon Footprint Management by Agricultural Practices”. The authors did a good effort but a few useful suggestions can improve the research write up. Few more general observations and some specific comments can be considered for the improvement of the MS before its acceptance for publication;
General Observation
· Give outlines of the MS.
· The basic objective of the MS is not given clearly in the abstract as it is given at the end of the introduction.
· Add recent literature (the year 2021 to 2022) with an emphasis on carbon footprint management under various agriculture practices scenarios.
· A paragraph should be at least 7 to 8 lines instead of 4 to 5 lines (e. g. Lines 115-119, 549-552, 553-556, 640-644) it may merge or extend.
· Try to follow a single format (Each word in upper case or in lower case) while writing headings and sub-headings (Line 150,217,260,275,350). Review all headings and sub-headings.
· The conclusion section should cover the objective and outline of the MS, improve this section.
· The references should be improved and updated according to the relevant journal format.
Some Specific Comments
· Line 53: Write the several methods which are used to determine CF estimation. Clear it.
· Line 56: You didn’t mention various issues by which the global food and agriculture industry is challenged. Clear this statement.
· Line 76: Write full form “million metric tons” instead of “mill metric tons”.
· Write full form of these symbols (CH4-Line 80; N2O-Line 88; ISO Line-108; FAO-Line 156) because these are not given before in full form in the MS.
· Line 90: You mentioned “In this country” mention the country name as well.
· Line 111: write available tools names and mention matching international standards….
· Use only the abbreviation “C” “SOC” and “GHGs” instead of the full form “carbon” (Line 170, 173, 176, 178, 186, 194…), “soil organic carbon” (Line 152, 322) and “greenhouse gases” (Line 218, 375) respectively or try to follow a single format. The full form of abbreviation might be used only for the first time. Review the whole MS and make possible changes.
· Line 171-172: Couldn’t understand the given statement rephrase it properly
· Line 181: Write only chemical formula “CH4” instead of full form “methane”
· Line 192-193: How can agricultural activities impact climate change due to direct and indirect GHGs emission? Clear it.
· Line 352: Discuss “chemical composition” in a few lines as well in this given 5.3. sub-heading
· Place Figure 2 in sub-heading 6.1. Inputs (Carbon sequestration) because fig. is discussed into this sub-heading
· Extend these (6.1.3. Biochar, 6.1.4. Crop residues, and 6.2.7. CO2 (tractors) harvesting and runoff) sections by adding a few more lines. These are given very short as compared to other sections.

Author Response
Reviewer 1
Respected editor, It is a great honor for me to review an MS entitled; “Carbon Footprint Management by Agricultural Practices”. The authors did a good effort but a few useful suggestions can improve the research write up. Few more general observations and some specific comments can be considered for the improvement of the MS before its acceptance for publication.
Response: We appreciate reviewers times and help to improve this paper. We have carefully placed changes pointed out by comments provided. Please find new version of the document.
General Observations
- Give outlines of the MS.
Response: We appreciate for the comment. We have added an outline to the MS. Please find below the abstract.
- Introduction
- Carbon footprint due to environmental factors
- Land-use changes and carbon footprint
- Agriculture and Carbon Footprint
- Role of soil in carbon footprint and agriculture
5.1. Soil Types
5.2. Soil health (Feedback mechanism)
5.3. Carbon Stabilization and Storage
- How does the system work?
6.1. Inputs (Carbon Sequestration)
6.1.1. Fertilizers
6.1.2. Manures Applications
6.1.3. Biochar
6.1.4. Crop Residues
6.1.5. Photosynthesis
6.2. Outputs (Carbon Emissions)
6.2.1. CO2 Emission (Soil Respiration)
6.2.2. CH4 Emission
6.2.3. N2O Emission
6.2.4. Carbon Leaching
6.2.5. Pesticides and Herbicides
6.2.6. Tillage
6.2.7. CO2 (Tractors), harvesting, and runoff
- Carbon footprint calculations
- Conclusion
- The basic objective of the MS is not given clearly in the abstract as it is given at the end of the introduction.
Response: Thank you for the comment. We have added a clear objective to the abstract. Please find on lines 23-25.
- Add recent literature (the year 2021 to 2022) with an emphasis on carbon footprint management under various agriculture practices scenarios.
Response: Thank you for the comment. We have added a recent publication.
- A paragraph should be at least 7 to 8 lines instead of 4 to 5 lines (e. g. Lines 115-119, 549-552, 553-556, 640-644) it may merge or extend.
Response: Thank you for the comment. We have either merged or extended aforementioned paragraphs.
- Try to follow a single format (Each word in upper case or in lower case) while writing headings and sub-headings (Line 150,217,260,275,350). Review all headings and sub-headings.
Response: Thank you for the comment. We have checked and updated all headings and subheadings. We used bolded regular headings, italic un-bolded subheadings, and regular un-bolded sub-subheadings.
- The conclusion section should cover the objective and outline of the MS, improve this section.
Response: Thank you for the comment. We have updated this section.
- The references should be improved and updated according to the relevant journal format.
Response: Thank you for the comment. We have updated this section.
Some Specific Comments
- Line 53: Write the several methods which are used to determine CF estimation. Clear it.
Response: Thank you for the comment. We definitely agree with reviewer how important to include CF estimation methods. Although, we originally write CF estimations in the paper, we removed it afterwards because the fact that not writing detail explanations mathematical estimation may not be enough to understand these methods correctly. If we add all description and explanations for CF estimations, this paper would be too long. We would like to keep that section for another review paper which may focus on CF estimations with meta-analysis examples.
- Line 56: You didn’t mention various issues by which the global food and agriculture industry is challenged. Clear this statement.
Response: Thank you for the comment. Since this paper only focus on agricultural management and their responsibility for CF, we do not want to dig into other industries. This may cause loss of focus for this specific paper.
- Line 76: Write full form “million metric tons” instead of “mill metric tons”.
Response: Thank you for the comment. We have corrected this.
- Write full form of these symbols (CH4-Line 80; N2O-Line 88; ISO Line-108; FAO-Line 156) because these are not given before in full form in the MS.
Response: Thank you for the comment. We have corrected this.
- Line 90: You mentioned “In this country” mention the country name as well.
Response: Thank you for the comment. We have corrected this.
- Line 111: write available tools names and mention matching international standards….
Response: Thank you for the comment. Since this paper only focus on agricultural management and their responsibility for CF, we do not want to dig into international standards etc. This may cause loss of focus for this specific paper since are a lot of international standards can go into this topic.
- Use only the abbreviation “C” “SOC” and “GHGs” instead of the full form “carbon” (Line 170, 173, 176, 178, 186, 194…), “soil organic carbon” (Line 152, 322) and “greenhouse gases” (Line 218, 375) respectively or try to follow a single format. The full form of abbreviation might be used only for the first time. Review the whole MS and make possible changes.
Response: Thank you for the comment. We have updated these parts.
- Line 171-172: Couldn’t understand the given statement rephrase it properly
Response: Thank you for the comment. We have updated this sentence.
- Line 181: Write only chemical formula “CH4” instead of full form “methane”
Response: Thank you for the comment. We have updated this.
- Line 192-193: How can agricultural activities impact climate change due to direct and indirect GHGs emission? Clear it.
Response: Thank you for the comment. We have updated this.
- Line 352: Discuss “chemical composition” in a few lines as well in this given 5.3. sub-heading
Response: Thank you for the comment. We have updated this part.
- Place Figure 2 in sub-heading 6.1. Inputs (Carbon sequestration) because fig. is discussed into this sub-heading
Response: Thank you for the comment. We have updated this part.
- Extend these (6.1.3. Biochar, 6.1.4. Crop residues, and 6.2.7. CO2 (tractors) harvesting and runoff) sections by adding a few more lines. These are given very short as compared to other sections.
Response: Thank you for the comment. We have updated these parts.
Reviewer 2 Report
This review article discussed carbon footprint management by environmental factors, land use, and agricultural practices. This review concluded that healthy soils have many benefits for a lower carbon footprint, and discussed the effect of tillage, diverse cropping systems, and machinery operations on soil structure. The topic is important to the field. The work is well organized and comprehensively described with relevant references for both historical literature and recent advances in the field. However, I suggest more discussion of existing gaps and future research directions so that readers can benefit more from this review article and thus potentially contribute to the continued development of the field.
Minor comments:
The Resolution of figure 2 is too low. The color of arrows and boxes should be clarified.
The font size is not consistent in line 532-544, line 764, and line 768.
Underscore is not needed in line 164.
Author Response
Response: We appreciate reviewers times and help to improve this paper. We have carefully placed changes pointed out by comments provided. Please find new version of the document.
Minor comments:
The Resolution of figure 2 is too low. The color of arrows and boxes should be clarified.
Response: We have created the figure 2 from the beginning. However, resolution have not changed so much. We are afraid the resolution is at highest possible from our end. Please find it on the paper.
The font size is not consistent in line 532-544, line 764, and line 768.
Response: We appreciate for the comments. The font size has been corrected. Please find line 532-768.
Underscore is not needed in line 164.
Response: We appreciate for the comments. The underscore has been removed. Please find line 164.